# Bioinformatics Designing and Molecular Modelling of a Universal mRNA Vaccine for SARS-CoV-2 Infection

**DOI:** 10.3390/vaccines10122107

**Published:** 2022-12-09

**Authors:** Elijah Kolawole Oladipo, Micheal Oluwafemi Adeniyi, Mercy Temiloluwa Ogunlowo, Boluwatife Ayobami Irewolede, Victoria Oluwapelumi Adekanola, Glory Samuel Oluseyi, Janet Abisola Omilola, Anietie Femi Udoh, Seun Elijah Olufemi, Daniel Adewole Adediran, Aanuoluwapo Olonade, Usman Abiodun Idowu, Olatunji M. Kolawole, Julius Kola Oloke, Helen Onyeaka

**Affiliations:** 1Genomics Unit, Helix Biogen Institute, Ogbomoso 210214, Nigeria; 2Laboratory of Molecular Biology, Immunology and Bioinformatics, Department of Microbiology, Adeleke University, Ede 231104, Nigeria; 3Department of Pure and Applied Biology, Ladoke Akintola University of Technology, Ogbomoso 210214, Nigeria; 4Department of Human and Clinical Anatomy, Ladoke Akintola University of Technology, Ogbomoso 210214, Nigeria; 5Department of Biochemistry, Obafemi Awolowo University, Ile-Ife 220005, Nigeria; 6Department of Biochemistry, Ladoke Akintola University of Technology, Ogbomoso 210214, Nigeria; 7Microbiology Unit, Department of Pure and Applied Biology, Ladoke Akintola University of Technology, Ogbomoso 210214, Nigeria; 8Department of Microbiology, University of Ilorin, Ilorin 234031, Nigeria; 9Department of Natural Science, Precious Cornerstone, Ibadan 200132, Nigeria; 10School of Chemical Engineering, University of Birmingham, Edgbaston, Birmingham B15 2TT, UK

**Keywords:** SARS-CoV-2, mRNA vaccine, bioinformatics, variants, lineages

## Abstract

At this present stage of COVID-19 re-emergence, designing an effective candidate vaccine for different variants of SARS-CoV-2 is a study worthy of consideration. This research used bioinformatics tools to design an mRNA vaccine that captures all the circulating variants and lineages of the virus in its construct. Sequences of these viruses were retrieved across the six continents and analyzed using different tools to screen for the preferable CD8^+^ T lymphocytes (CTL), CD4^+^ T lymphocytes (HTL), and B-cell epitopes. These epitopes were used to design the vaccine. In addition, several other co-translational residues were added to the construct of an mRNA vaccine whose molecular weight is 285.29686 kDa with an estimated pI of 9.2 and has no cross affinity with the human genome with an estimated over 68% to cover the world population. It is relatively stable, with minimal deformability in its interaction with the human innate immune receptor, which includes TLR 3 and TLR 9. The overall result has proven that the designed candidate vaccine is capable of modulating cell-mediated immune responses by activating the actions of CD4^+^ T cells, natural killer cells, and macrophages, and displayed an increased memory T cell and B cell activities, which may further be validated via in vivo and in vitro techniques.

## 1. Introduction

On 30 January 2020, the World Health Organization made the declaration on SARS-CoV-2 a public health emergency of global impact, and since this time, biotechnology companies have utilized various technologies to develop vaccines for the virus. Among the developed vaccines are Pfizer, an mRNA vaccine developed by BioNTech; AstraZeneca, a non-replicating viral vector developed by Oxford University; and BBV152 COVAXIN, developed by Bharat Biotech through inactivated virus technology [1]. As of September 2022, there were 36 vaccine candidates under clinical trial and 146 in preclinical evaluations (WHO, 2020) [2]. However, there exists a global equity challenge in manufacturing a candidate vaccine for SARS-CoV-2 [3]. This challenge might have surfaced as a result of developing a vaccine based on short-term data at the initial stage of the pandemic [4] and also the rapid requirement for a vaccine for the pandemic in an unprecedented time frame during these initial periods of the pandemic, which allowed the omission of the initial step of exploratory vaccine design and development at an earlier stage [5]. With the re-emergence of new SARS-CoV-2 virus variants around the world, the induction of immune responses and efficacies of existing vaccines are bound to reduce, especially with the recent fallbacks in the currently available vaccine and anomalies reported with the available vaccines. We opt to design an mRNA vaccine that will solve the problem of equity in variant representation, and that will be effective across all the continents of the world.

The choice of an mRNA virus was informed by its advantages over DNA vaccines subunit, killed, and live attenuated vaccines. Due to the progressive translation of mRNA vaccines into encoded proteins and peptides to activate long-lasting effects better than a peptide-based vaccine, it also possesses a wide range of potential for correcting abnormalities such as protein expression and higher therapeutic efficacy [6]. Therefore, we sought to design and model an mRNA vaccine that will elicit an immune response across a wide range of people considering that we integrated about twenty-one circulating lineages (21) of SARS-CoV-2, which were representatives for the analysis and carefully selected across all the continents.

## 2. Materials and Method

### 2.1. Study Outline

This study followed a structured workflow, which is illustrated in Figure 1. The structure highlighted all the steps followed in designing this universal mRNA vaccine.

### 2.2. Retrieval of Whole Genome Sequence of Sar-Cov-2 Variants

Representative genomic sequences for 21 lineages of SARS-CoV-2 from Africa, Asia, Australia, Europe, North America, and South America have been retrieved from the Global Initiative for Sharing All Influenza Data (GISAID) database for further analysis. These variants include Alpha, Beta, Gamma, Omicron, and their sub-variants [7]. The sequences were mapped out with the spike glycoprotein of the Wuhan SARS-CoV-2 reference sequence, whose ascension number is NC_045512.

### 2.3. MHC I and MHC II Binding Epitopes Prediction

In designing this vaccine, choosing epitopes that are potentially immune-protective comes first. The epitopes binding affinity to be used in vaccine design, with the major histocompatibility complex (MHC) allele, is an important criterion that must be considered [8,9]. The annotated whole genome sequences were predicted for their MHC I and MHC II binding affinity to select the CTL and HTL epitopes used for this vaccine construct, respectively. NetCTL 1.2 tool was used for CTL epitope prediction, while IEDB MHC II servers were used for HTL prediction [10,11]. The NetCTL server was used to predict the MHC class I supertype A with a 0.75 epitope identification threshold, while the IEDB MHC II immunoinformatics tool predicted from its database epitopes that have binding affinity to the reference set of human alleles.

### 2.4. Prediction and Evaluation of Linear B-Cells (LBL) Epitopes

B-cells are essential in the provision of long-term protection against pathogens and antigens [12]. In an epitope-driven vaccine, the deduction of B-cell epitopes from the target antigen is crucial [13]. The B-cell epitopes in the annotated SARS-CoV-2 genome were predicted using two different immunoinformatics tools, BCPred and ABCPred, and the most reoccurring results were selected [14,15].

### 2.5. Evaluation of Cross-Reactivity of Selected Epitopes

Memory T cells formed by past viral antigens to which humans have been exposed have the ability to recognize and cross-react with host cell proteins [16]. Consequently, all eleven (11) epitopes that were finally selected to be used to construct this mRNA vaccine were screened for their presence in the human genome using the NCBI Basic Local Alignment Search Tools (BLAST).

### 2.6. Population Coverage Prediction

The two classes of leukocyte antigens specific to humans, HLA I and II, are primarily associated with CTL and HTL, which are most important for antiviral immunity responses and regulation of antibody and cytotoxic responses, respectively, in the human immune system. The population coverage prediction of the epitopes that bind to the MHC I and MHC II molecules [17] was carried out by the IEDB Population Coverage tool to determine the population size that would elicit an immune response to the constructed vaccine.

### 2.7. In Silico Vaccine Construction

In constructing the vaccine, the selected epitopes and some other co-translational residues were added in an order used by Oluwagbemi et al., 2022 [18] in designing an mRNA vaccine against SARS-CoV-2. The 5′ cap, which comes first, ensures stability and functionality in mRNA vaccines. The 5′ UTR that followed helps in the regulation of translation and mRNA stability. The Kozak sequence contains the start codon. Human Beta Defensin (HBD1), a TLR3 agonist, is used as an adjuvant to co-stimulate the ability of the vaccine to trigger an antigenic reaction [19]. Additionally, the MHC II binding epitopes and the adjuvant were looped by GPGPG linkers, (EAAK)2 linked the MHC II binding epitopes with the LBL epitopes, LBL epitopes were interspaced with (EAAK)2, and LBL and MHC I binding epitopes were linked by AAY.

A signal peptide, tissue plasminogen activator (tPA), and an MHC I-targeting domain (MITD) were added. The tPA is involved in the egression of translated epitopes to the circulatory system, and the MITD helps direct CTL epitopes to the MHC-I compartment of the endoplasmic reticulum. At the latter end of the construct, the 3′ UTR and PolyA tail were added, also for the purpose of improving the mRNA stability and translation.

### 2.8. Prediction of Antigenicity, Allergenicity, Toxicity and Physicochemical Properties

After the construction, the candidate vaccine’s antigenicity, allergenicity, and toxicity were determined using different bioinformatics tools. VaxiJen, an antigenicity prediction tool, was used [20]. The probability of the vaccine causing an allergic reaction was also predicted using AllerTop [21]. As toxicity is one of the limitations faced with using protein/peptide-based therapy, the toxicity of the vaccine was accessed using ToxinPred2 [22].

### 2.9. Secondary Structure Prediction

The secondary structure of the vaccine, which shows the amino acid interactions, is an essential step as it offers information about the amino acid relationship in the construct [23]. SOPMA, a tool that offers notable improvements in the mRNA vaccine’s secondary structure prediction by using similar predictions from multiple alignments [24], was employed.

### 2.10. Tertiary Structure Prediction, Conformational BCELL Epitopes Prediction and Ramachandran Plot

The 3D model structure of the vaccine construct was predicted using Phyre2 [25]. This bioinformatics tool employed advanced backend remote correlation detection methods to assemble 3D models of the protein, predict ligand binding sites, and also analyse the effect of amino acid variants (e.g., nonsynonymous SNPs (nsSNPs)) in the protein sequence submitted. Both linear and conformational B-cell epitope are known to be antigenic determinants, or antigen-binding portions of B-cells. Therefore, the tertiary structure was subjected to in silico detection of conformational B-cell epitopes using ElliPro, a method that uses a structure-based tool for the deduction of antibody epitopes [26]. The Ramachandran plot, which shows amino acids within the energetically favouring region, was also predicted using ProCheck [27].

### 2.11. Molecular Docking of Vaccine-TLRs

Toll-Like Receptors (TLRs) are proteins that recognize distinct patterns of antigenic molecules derived from microorganisms and induce an immune response culminating in the secretion of pro-inflammatory cytokines and type I interferons (IFNs) [28]. TLR 3 and TLR 9 receptors have been reported to recognize microbial or viral nucleic acids [28,29]. Hex 8.0.0 software was used to dock the tertiary structure of the vaccine construct against TLR3 and TLR9 retrieved from AphaPhod, i.e., ID Q1KMK2 and B4E0A1, respectively [30].

### 2.12. Molecular Dynamics Simulation Analysis

Molecular dynamics simulation is a computational technique that captures the detailed atomic behaviour of proteins at a good temporal [31]. In order to obtain the interaction between the vaccine and the TLR 3 and TLR 9 complexes, the two complexes were subjected to IMOD, a tool that explores normal mode analysis (NMA) in internal (dihedral) coordinates that naturally reproduce the joint functional motions of biological macromolecules and generates feasible transition pathways between two homologous structures, even with large macromolecules [32].

### 2.13. Immune Response Simulation

The recognition of antigenic peptides, which are the epitopes used in this vaccine construct, is essential to the immunogenicity of the vaccine. The C-immsim tool was used to predict the immune response simulation of the vaccine [33]. The criteria for immune response simulation on the server were set to default for the prediction.

## 3. Result

### 3.1. Prediction and Analytical Evaluation of MHC Class I and II Binding Epitopes

HTL and CTL activation require binding epitopes in the specialized groove of MHC molecules; the CTL epitopes bind with MHC class I, while the HTL epitopes bind with MHC class II. The epitopes were predicted for all the antigenic sequences retrieved for the design of the universal mRNA vaccine candidate, which includes all the representative sequences from available continents.

Prediction of the CTL epitopes was carried out using the 12 MHC class I supertypes from the server as a default [10]. The CTL epitopes predicted were 44 in total, but only 14% of the epitopes passed the analysis carried out on them, which includes the antigenicity, toxicity, and allergenicity as shown in Table 1 below.

Out of the 116 epitopes predicted using the MHC class II allele tool, 52% of HTL is antigenic, non-allergenic, and non-toxic. However, only two epitopes were found to be IFN-γ inducers: IL-10 and IL-4 inducers, as depicted in Table 2 below. The antigenicity scores of the two are 1.2362 and 0.5881, respectively, which passed the antigenicity test that has a threshold of ≥0.5. One of the epitopes bounded with 17 HLA Class II alleles; therefore, it was selected as the final HTL epitope for the mRNA vaccine construction.

### 3.2. Prediction and Evaluation of Linear B Cells (LBL) Epitopes

A total of 27 B-cell epitopes were deduced from the antigenic sequences, and 15% of the retrieved epitopes passed the antigenicity, toxicity, and allergenicity tests before they were subjected to these predictions. The epitopes selected from the antigenicity score are those higher than the threshold of 0.5, as shown in Table 3 below.

### 3.3. Population Coverage

The results showed that Europe had a population coverage score of 77.07%, Central Africa had a score of 52.98%, Central America had a score of 5.69%, East Africa had a score of 57.44%, East Asia had a score of 55.06%, North Africa had a score of 60.89%, North America had a score of 66.7%, Northeast Asia had a score of 65.94%, Oceania had a score of 54.1%, South Africa had a score of 61.29%, and finally the world had a score of 68.87%.

As a result, Central America had the lowest population coverage percentage score, at 5.69%, and Europe had the highest, totaling 77.07% as depicted in Figure 2 below. This demonstrates that the pandemic will be effectively fought off by the vaccination developed using these chosen epitopes.

### 3.4. Evaluation of Cross Affinity

Right before the mRNA universal vaccine construction for SARS-CoV-2, the epitopes chosen for construction were blasted against the *Homo sapiens* database resources on NCBI using NCBI BLAST. This procedure helped in determining that the epitopes do not have a cross-affinity with any gene expressed in humans to prevent a cross-reactivity when introduced into the human body system. The result indicated that none of the predicted epitopes can be found in the human genome; therefore, there will not be a cross-reaction in humans.

### 3.5. mRNA Universal Vaccine Construction for SARS-CoV-2

A total of eleven epitopes were selected for the vaccine candidate after carrying out different analyses on them. These epitopes include 4 B-cell epitopes, 6 CTL epitopes, and an HTL epitope. Other residues added in the construction of the primary construct include the 5′ cap, 5′ UTR, Kozak sequence, and tPA (signal peptide), which were combined with the adjuvant (β defensin 1) and then joined to the HTL epitope with the aid of the GPGPG linkers to form the universal mRNA vaccines as depicted in Figure 3 below. Epitopes were sequentially linked using the linkers EAAKEAAK (HTL to LBL) and AAY (LBL to CTL and intra-CTL), respectively, depending on how compatible their interactions were [18]. The other terminal end of the constructed CTL epitopes was linked together by AAY linkers.

### 3.6. Prediction and Evaluation of the mRNA Vaccine’s Secondary Structure

The secondary structure of the vaccine construct was predicted and analyzed employing the SOPMA server [24]. The result of the analysis revealed a stabilized structure for the vaccine construct with 31.38% alpha-helix, 18.11% extended strands, 18.11% beta-turn, and 41.10% random coils in Figure 4. This outcome also demonstrated that the secondary structure of the vaccine construct had high globular conformation, flexibility, and stability.

### 3.7. 3D Structure Modelling and Evaluations

The mRNA vaccine candidate’s tertiary structure was modelled using Phyre2 as shown in Figure 5. For the evaluations of the 3D structure, five conformational B-Cells epitopes were predicted, as shown in Figure 6 [25,26], Ramachandran plot score statistics revealed that 87.6% of the amino acids residues are on the favored region and 11.0% in the additional allowed region, as shown in Figure 7. Results from predicted physicochemical properties illustrate that the vaccine construct is made up of 2540 amino acids and has a 285.29686 kDa molecular weight. The mRNA vaccine construct was predicted to be slightly basic in nature, with an estimated pI of 9.20. The half-life predicted for the vaccine is approximately 30 h in mammalian reticulocytes in vitro. The instability index of the vaccine was 48.74, which is above the threshold value of 40, emphasizing the instability of mRNA. The aliphatic index of the construct was 65.85, which indicates the construct has been thermostable [34]. The hydropathicity nature of the mRNA construct was proven by its GRAVY score of −0.604, which is a negative value [35].

### 3.8. Molecular Docking of Vaccine with TLRs

The evaluation of the interaction between the vaccine candidate and potential receptor was conducted through molecular docking, utilizing the Hex 8.0.0 software. This procedure is performed to determine and verify the binding affinity between the complexes (vaccine-TLRs) [18]. Protein–protein docking carried out between the final refined 3D vaccine and the TLR3 and TLR9 immune receptors is shown in the diagram below. A blind molecular docking procedure was utilized in the process, leaving the TLRs to bind with the vaccine construct at the most favourable site, as shown in Figure 8 below.

### 3.9. Molecular Dynamics Simulation

The molecular dynamic simulation of the mRNA vaccine construct with TLR3 and TLR9 predicted by the IMOD server shows the result of the spin structure predicted for the Ligand–Receptor Interaction, the deformability graph depicting a little deflection in the 0 to 1 °A range in the coordinates, the B-factor mobility depicting the atomic displacements of the proteins within the range of 0 to 1 °A, the NMA mobility eigenvalues, which are 1.410683 × 10^−7^, the elastic network, and the covariance matrix analysis, which showed the atomic pairs of the complexes. Figure 9 below revealed correlated portions in the red color, non-correlated portions in the blue color, and uncorrelated portions in the white color [36].

### 3.10. Immune Response Simulation

The immune simulation works on a model that shows the interaction and activations of cells in the innate and adaptive systems. The C-Immsim results in Figure 10 show that dendritic cells, macrophages, and epithelial cells are activated and are capable of presenting antigens in the specialized groove of MHC-I and MHC-II molecules on a single dose of the multi-epitope vaccine construct [33]. It also shows that CD4 T-helper lymphocytes, CD8+ lymphocytes, and NK cells were predicted to be moderately activated within the first few days of exposure to the designed vaccine. B lymphocytes and plasma B lymphocytes were also predicted to get activated and secrete IgM and IgG isotypes, and the secretion of cytokines is enhanced when the candidate vaccine is administered.

## 4. Discussion

Following the outbreak of SARS-CoV-2 in 2019, which has posed several threats to world health, ditto to this, several therapeutics have been developed to further curb the threat. Invariably, due to the nature of the virus, several variants and lineages have been reported after several vaccines were developed [37]. We designed the mRNA vaccine utilizing immunoformatics techniques, and the CTL and HTL epitopes used in the designed vaccine were predicted from the SARS-CoV-2 spike glycoproteins. The CTL-binding epitopes integrated into the vaccine are recognized by human CTL through their binding to the MHC class I alleles. They appear in the CTL as fragments of viral proteins that have undergone degradation and thereby release cytotoxic granules that kill infected cells [38].

To induce a significant antibody response, the purified antigens had to be administered in sufficient doses, with the fact that quite a respectable number of antibody and T-cell responses are generated after vaccination [39]. The selected epitopes and some other compounds were added in accordance with Oluwagbemi et al., [18], in designing an mRNA vaccine against SARS-CoV-2 in 2020. The stability of the designed mRNA and its functionality were achieved by the presence of the 5′ cap and the 5′ UTR, and the Kozak sequence added conferred on the mRNA vaccine construct the functionality of initiating the process of translation in the eukaryotic cells [40]. Adjuvants are necessary to aid a vaccine’s immunogenicity, effectiveness, efficacy, and half-life [41]. In the designed vaccine, Human Beta Defensin (HBD1), a TLR3 agonist was used as an adjuvant to co-stimulate the ability of the vaccine to trigger an antigenic reaction [19].

A plasminogen activator, tPA, was added to complement the egress properties of the translated mRNA out of the host immune cells, and also MITD conjugated with the designed mRNA helps to signal the CTL epitopes to the MHC-I segment of the endoplasmic reticulum and its easy presentation to the surface of antigen presentation cells. The 3′ UTR and the polyA tail added to the 3′ ends of the designed mRNA also help with stability and translation initiation [42].

Cross-reactivity in the selected epitopes, after it was screened against the human genome, ascertain its safety characteristics and non-insertional mutagenesis.

The population coverage of the constructed vaccines covers about an average of 68.87% of the entire world population, showing that the construct has the ability to induce immune responses in over 75.94% of Europe and 69.12% of North America, which is similar to work conducted by Ahammad and Lira, [43].

The vaccine construct’s physicochemical characteristics demonstrate its suitability as a promising prospect for a vaccine. The molecular weight (MW) of the mRNA vaccine design is 285.3 kDa, and the estimated theoretical pI is 9.20, indicating the vaccine is fairly basic, which is in the same range as a similar vaccine designed Ahammad and Lira, [43]. The instability index’s result is 48.74, which implies that the protein would be fairly stable. The aliphatic index of the construct is 65.85, which signifies that the construct is thermostable [34]. The hydropathicity nature of the mRNA construct was proven by its GRAVY score of −0.604 [35].

Analysis of the secondary structure of our vaccine showed that the designed vaccine’s secondary structure is stable, has good flexibility, and has globular conformation.

In addition to the good conformation and stability expressed in the result of secondary structure, the map of allowed and disallowed conformation revealed by the Ramachandran plot depicts that the vaccine-construct amino acid is in the most favoured region with a coverage of 87.6% [27,44].

The vaccine construct has a high binding affinity for TLR 3 and TLR 9, which had been reported by Takeuchi in 2010 [28], as it has a binding score of −143.99 for TLR3 and 0 for TLR9. This low binding energy score indicates strong affinities between the molecules [45].

Prediction of atomic stability of the vaccine with TLR in silico was carried out imploring the molecular dynamics simulation technique, and our corresponding result shows it was favourable when compared with Oluwagbemi et al., [18]. The induction of an immune response shown by the in silico immune simulation indicates that appropriate antibodies were secreted upon administration of the antigen.

The designed mRNA vaccine has shown to likely be a promising candidate against all circulating variants of SARS-CoV-2, and therefore further in vivo and in vitro analysis needs to be carried out for validation of it efficiency and potency in inducing immune response against the virus.

## 5. Conclusions

Due to the ever-mutating character of SARS-CoV-2, this study has designed and modelled a potential vaccine candidate integrating the circulating variants and about twenty-one (21) SARS-CoV-2 lineages, which in-silico techniques have ascertained to be antigenic, non-toxic, and non-allergic, which may be further evaluated via in-vivo studies.

## Figures and Tables

**Figure 1 vaccines-10-02107-f001:**
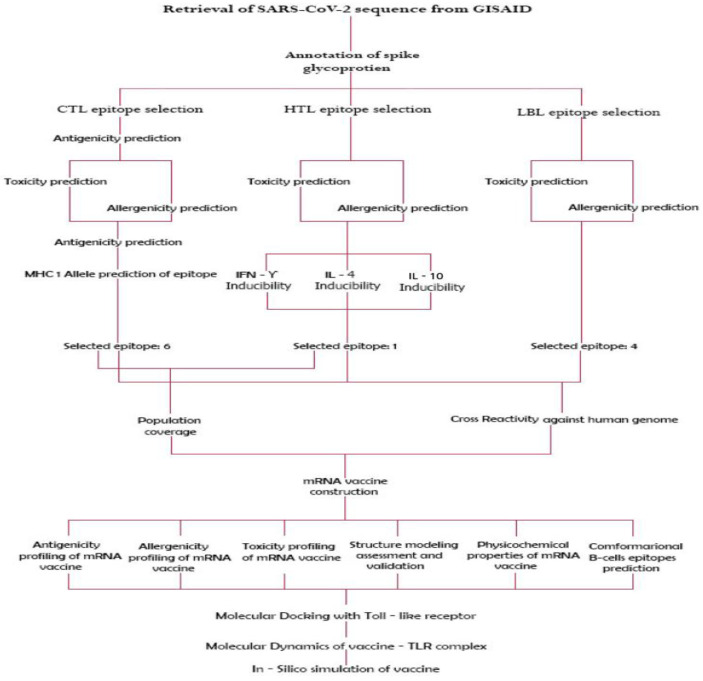
Workflow for the universal mRNA vaccine construct.

**Figure 2 vaccines-10-02107-f002:**
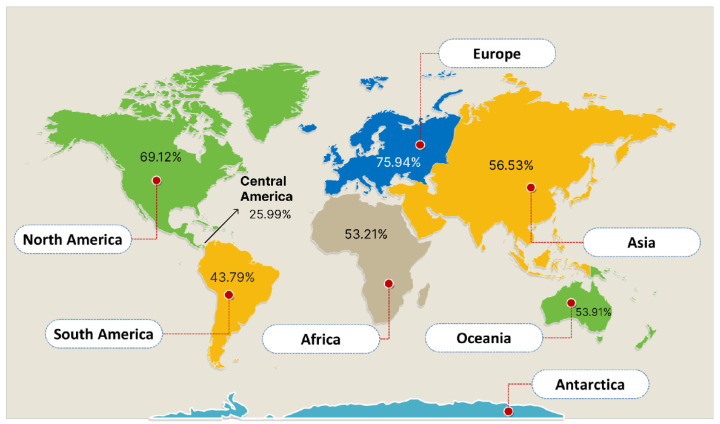
Prediction of population coverage of the selected T cell epitopes.

**Figure 3 vaccines-10-02107-f003:**
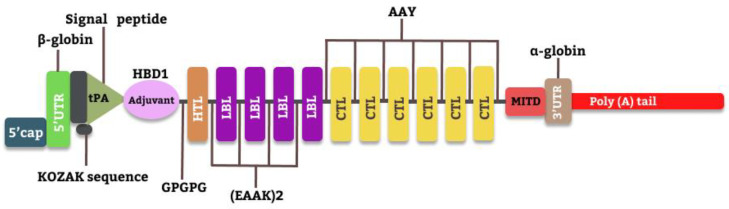
The schematic representation of the mRNA universal construct.

**Figure 4 vaccines-10-02107-f004:**
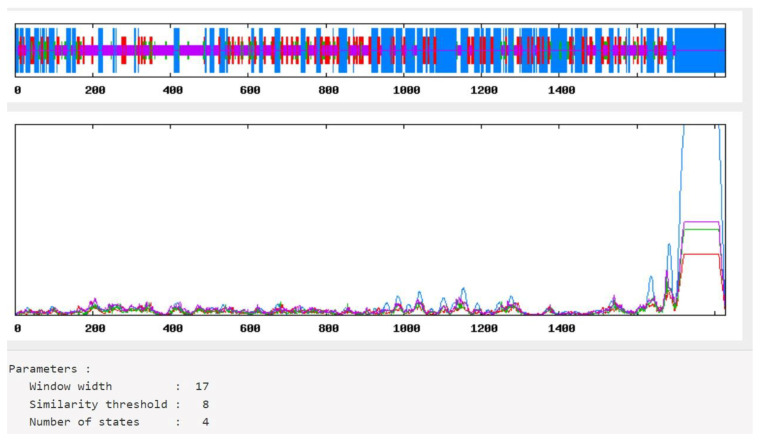
Secondary structure prediction of vaccine construct.

**Figure 5 vaccines-10-02107-f005:**
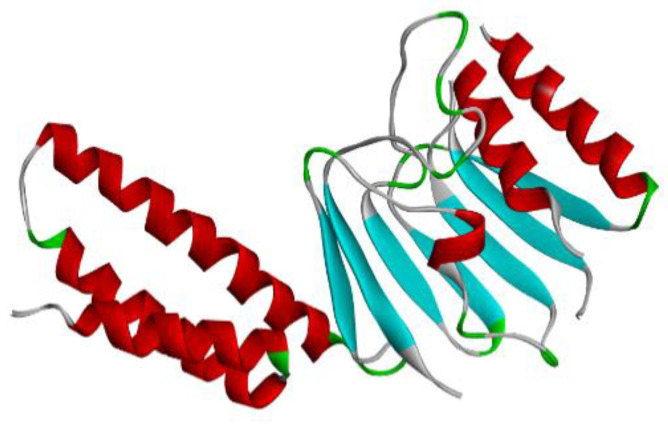
3D model of the vaccine construct.

**Figure 6 vaccines-10-02107-f006:**
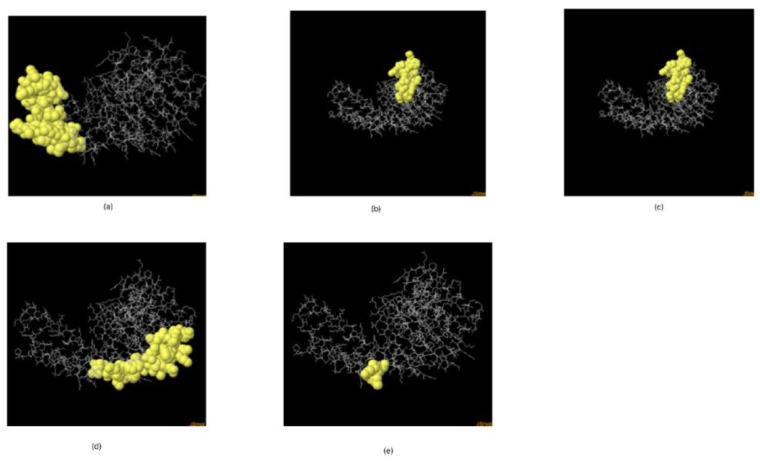
The 3D model of the 5 predicted conformational B-cell epitopes. The yellow regions are the conformational B-cell epitopes, while the grey regions are the residue remnant. (**a**) 21 residues has a score of 0.915; (**b**) 40 residues has a score of 0.758; (**c**) 46 residues has a score of 0.702; (**d**) 11 residues has a score of 0.674; (**e**) 6 residues has a score of 0.659.

**Figure 7 vaccines-10-02107-f007:**
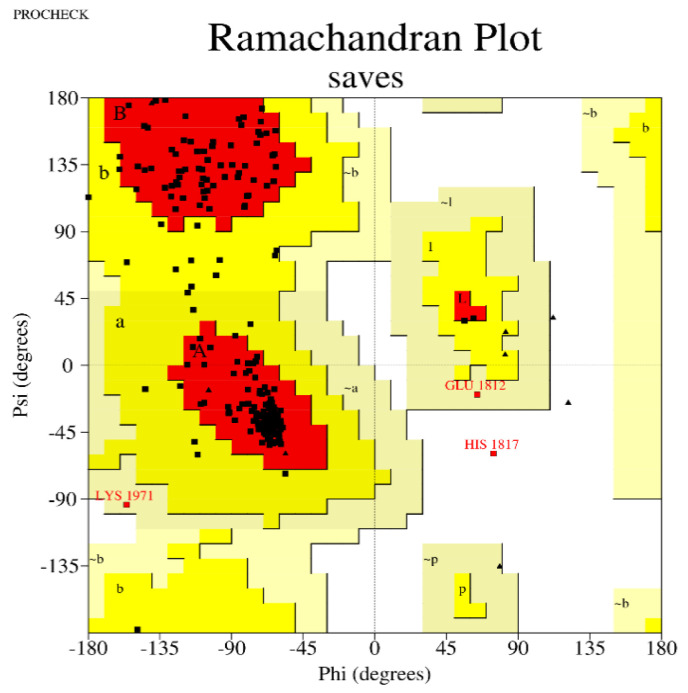
Ramachandran residues score plot.

**Figure 8 vaccines-10-02107-f008:**
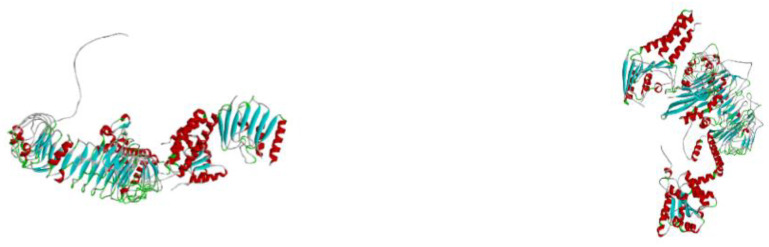
3D model of vaccine-TLR3 and TLR9 complexes.

**Figure 9 vaccines-10-02107-f009:**
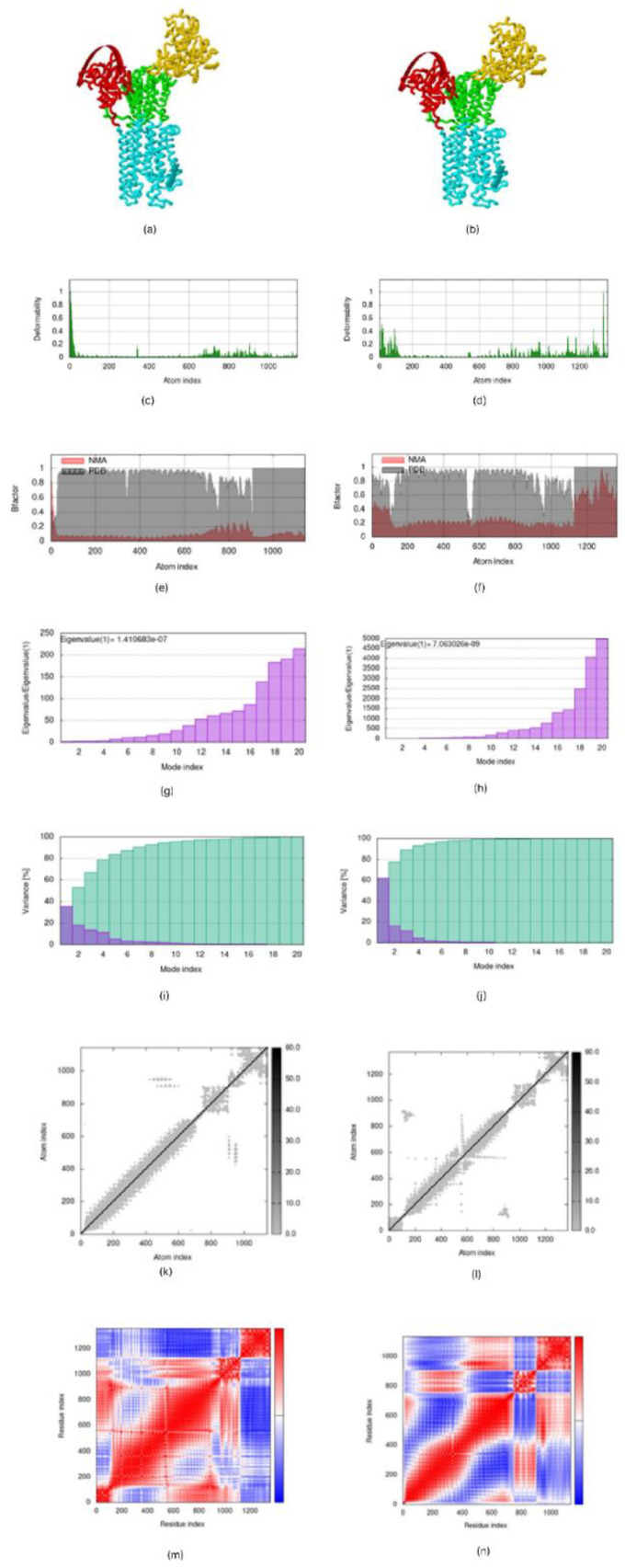
Molecular dynamic simulation results for vaccine-TLR3 and TLR9 complexes showing (**a**,**b**) spin predictions of the Ligand-Receptor Interaction (**c**,**d**) deformability (**e**,**f**) B-factor mobility (**g**,**h**) NMA mobility eigenvalues (**i**,**j**) variance (**k**,**l**) elastic network analysis (**m**,**n**) covariance matrix.

**Figure 10 vaccines-10-02107-f010:**
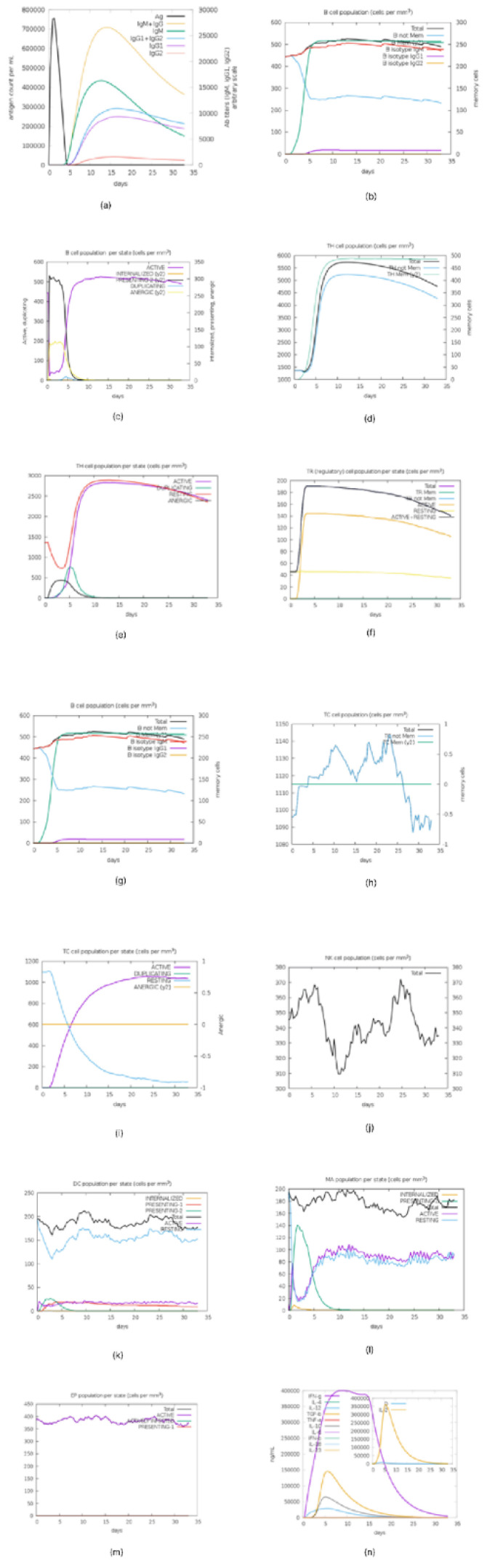
Immune simulation prediction: (**a**) Antigens and immunoglobulins. Antibodies are sub-divided per isotype. (**b**) Lymphocytes B total count memory cells. (**c**) B lymphocytes population per entity-state (i.e., showing counts for active, presenting on class-II, internalized the Ag, duplicating and anergic. (**d**) CD4 T-helper lymphocytes count. Total and memory counts were shown on the plot. (**e**) CD4 T-helper lymphocytes count sub-divided per entity-state (i.e., active, resting, anergic, and duplicating). (**f**) Plasma B lymphocytes count sub-divided per isotype (**g**) CD4 T-regulatory lymphocytes count. (**h**) CD8 T-cytotoxic lymphocytes count. Total and memory shown. (**i**) CD8 T-cytotoxic lymphocytes count per entity-state (**j**) Natural killer cells (total count). (**k**) Dendritic cells. (**l**) Macrophages. Total count, internalized, presenting on MHC class-II, active and resting macrophages. (**m**) Epithelial cells. (**n**) Cytokines (concentration of cytokines and interleukins with D signifying danger.

**Table 1 vaccines-10-02107-t001:** The HTL epitopes selected after screening through the toxicity, allergenicity, antigenicity, IFN-Y, IL-4, and 1L-10 predictions.

Epitopes	Toxicity	Allergenicity	Antigenicity	IFN-γ	IL-4	IL-10
LFCFHEVHNKRLDFW	non-toxic	non-allergic	antigenic	inducer	inducer	inducer
PILVQFQVFMISFHV	non-toxic	non-allergic	antigenic	inducer	inducer	inducer

**Table 2 vaccines-10-02107-t002:** The CTL epitopes selected after screening through toxicity, allergenicity, and antigenicity predictions.

CTL Epitopes	Toxicity	Allergenicity	Antigenicity
LTLITLLPY	non-toxic	non-allergic	antigenic
LTSLGFKLY	non-toxic	non-allergic	antigenic
STTHMSVTY	non-toxic	non-allergic	antigenic
DTDTSLTPF	non-toxic	non-allergic	antigenic
LVQVLHYKY	non-toxic	non-allergic	antigenic
ATSLSVCFY	non-toxic	non-allergic	antigenic

**Table 3 vaccines-10-02107-t003:** The B cell epitopes selected after screening for toxicity, allergenicity, and antigenicity predictions.

B-Cell Epitopes	Toxicity	Allergenicity	Antigenicity
AFSYGPRKTGFQKSGICVEY	Non-Toxin	Probable Non-Allergen	0.9349
GLNMSTTHMSVTYPLVQVYA	Non-Toxin	Probable Non-Allergen	0.5762
MGHFAWWTAFVTNVNASSSE	Non-Toxin	Probable Non-Allergen	0.5311
HFWMFITTKTTKVGKVSSEF	Non-Toxin	Probable Non-Allergen	1.0468

## Data Availability

All data analyzed in this study are publicly available data at the National Centre for Biotechnology Information GenBank repositories and the Global Initiative for Sharing All Influenza Data (GISAID), which could be made available upon request.

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
