# Peer review of "Bioinformatics Designing and Molecular Modelling of a Universal mRNA Vaccine for SARS-CoV-2 Infection"

_vaccines, 2022, doi:10.3390/vaccines10122107_

Round 1

Reviewer 1 Report

Line 26: “continent” replace with “continents”

Line 29:  “co-translational residues were added”. What do you mean by r co-translational residues?

Line 30: estimated pI was 9.2, which would not have any cross affinity.

                How you are sure that there is no cross reactivity? Did you verify that?

The last sentence of abstract: it is overstated. Without experimental validation how the authors conclude that the vaccine is capable of modulating immune response. Just based on the Insilco experiments they can’t say that.

Introduction needs details explanation about the problem of the study, approches and study reports.

Methods:

Line 69: 477 whole genomes were retrieved. What is the basis to select these 477 in spite of availability of thousands of genomes in GISAID?

Line 74: sequences were annotated with the spike glycoprotein of SARS-CoV-2 reference sequence

Annotation means comparison here? Annotation gives a different meaning which doesn’t opt in this sentence.

If interested in only spike glycoprotein, why whole genome sequences were retrieved? Why not only spike glycoprotein seqs?

What are the inputs you have provided to the tools? Whole genome sequences? If yes, you cant as they accept only protein sequences. If protein sequences, which proteins again. Methods part is confusing to understand. Please state the clear methods, so that any general audience can understand.

Results:

Investigation of Cross Affinity: method was explained. But authors didn’t show the result here.

Figure 7: please describe the statistics of Ramachandran plot with the respect to the quality of structure.

MD results: please explain the main parameters such as kinetic energy and RMSD.

Author Response

Reviewer 1 comments and suggestions :

Comments and Suggestions for Authors

Comments: Line 26: “continent” replace with “continents”

Responses: The comment has been duly noted and adjusted in the manuscript

Comments: Line 29:  “co-translational residues were added”. What do you mean by r co-translational residues?

Response: The co-translational residues i.e kozak sequences added to the mRNA construct has been reported from literature and research works to serves as initiation site for the mRNA expression at optimal rate.

Comments: Line 30: Estimated pI was 9.2, which would not have any cross affinity. How you are sure that there is no cross reactivity? Did you verify that?

Response: Computationally, we determine if there is no cross reactivity of the selected epitopes with the human sequences to avoid inflammation. The cross reactivity was predicted by blasting the epitopes against the Homo sapiens database resources before construction. The epitopes shows not to have any relation with Homo sapiens genome, therefore it could act sufficiently as an antigen.

Comments: The last sentence of abstract: it is overstated. Without experimental validation how the authors conclude that the vaccine is capable of modulating immune response. Just based on the Insilco experiments they can’t say that.

Response: Based on Insilico analysis, we were able to determine the immune simulation of the potential vaccine candidate. However, this has not been verified via experimental analysis and thereby acknowledge this comment and therefore modify the statements to best indicate our intentions.

Comments: Introduction needs details explanation about the problem of the study, approaches and study reports.

Response: The comment has been duly noted and adjusted in the manuscript

Methods:  

Comments: Line 69: 477 whole genomes were retrieved. What is the basis to select these 477 in spite of availability of thousands of genomes in GISAID?

Response: Representative sequences were selected across the all the continents in all circulating lineages and variant across the world whereby sequences where selected based on their completed metadata information, where all those with completed information were used for further analysis which were in total of 477.

Comments: Line 74: sequences were annotated with the spike glycoprotein of SARS-CoV-2 reference sequence. Annotation means comparison here? Annotation gives a different meaning which doesn’t opt in this sentence.

Response: Although, Annotation means comparison, in the context of our research work we compared the retrieved sequences with the reference dataset to map out the region of interest in the sequence. Another word which best suite this analysis has been updated in the manuscript.

Comments: If interested in only spike glycoprotein, why whole genome sequences were retrieved? Why not only spike glycoprotein seqs?

Response: The GISAIDs database used to access these genomic information only allows the retrieval of whole genome sequences therefore mapping out the needed region is essential for the analysis to proceed.  

Comments: What are the inputs you have provided to the tools? Whole genome sequences? If yes, you can’t as they accept only protein sequences. If protein sequences, which proteins again. Methods part is confusing to understand. Please state the clear methods, so that any general audience can understand.

Response: The method adopted in this research work have been widely reported in literature, in the manuscript we have made more clarifications where necessary, in order to aid the understanding of our intended audiences 

Results:

Comments: Investigation of Cross Affinity: method was explained. But authors didn’t show the result here.

Response: The “Investigation of Cross Affinity” section was included in the result section, however, more information has been included to further explain our findings.

Comments: Figure 7: please describe the statistics of Ramachandran plot with the respect to the quality of structure.

Response: The comment has been duly noted and has been improved upon in the manuscript

Comments: MD results: please explain the main parameters such as kinetic energy and RMSD.

Response: The Normal mode analysis, a computational method employed by Imod suite for the computation of the oscillation, motion and vibrations of macromolecules relationship and due to the fact that the construct has large molecular size which are effectively analysed by Normal mode analysis.

Reviewer 2 Report

Dear Editors

I have revise the article: Bioinformatics Designing and Molecular Modelling of a Uni- 2 versal mRNA Vaccine for SARS-CoV-2 Infection by Elijah Kolawole Oladipo et al. It is an interesting work regarding to design mRNA vaccines under in silico studies. However, I have some suggestions:

1.- I suggest to authors describe the sequence analyses to select the consensus sequence for further studies.

2.- It is no clear whether sequence analysed are from RNA or  proteins 

3.- I suggest to consider the receptors from lymphocyte and explain the Toll Like Receptors

4.- I suggest to show the sequences of predicted peptides due to several have been reported elsewhere

5.- I suggest to perform an experimental assays of the most promissory peptide for in silico validation

Author Response

Reviewer 2 comments and suggestions :

I have revise the article: Bioinformatics Designing and Molecular Modelling of a Uni- 2 versal mRNA Vaccine for SARS-CoV-2 Infection by Elijah Kolawole Oladipo et al. It is an interesting work regarding to design mRNA vaccines under in silico studies. However, I have some suggestions:

Comments1.- I suggest to authors describe the sequence analyses to select the consensus sequence for further studies.

Response: This suggestion is duly acknowledged

Comments 2.- It is no clear whether sequence analysed are from RNA or  proteins 

Response: The end product of the research is to design universal mRNA vaccine, however for some post-construction analysis, the RNA sequence must be converted to proteins.           

Comments 3.- I suggest to consider the receptors from lymphocyte and explain the Toll Like Receptors

Response: This suggestion is duly acknowledged, however the epitopes binding affinity to the MHC-II molecules has been predicted, which would in turn helps activates the lymphocytes via it presentation to it receptor (T-cell receptor).

Comments 4.- I suggest to show the sequences of predicted peptides due to several have been reported elsewhere

Response: This suggestion is duly acknowledged, as the research work is intended to be a translational product, the authors have decided to show the peptide sequence in the next phase of the work were the vaccine would be evaluated.

Comments 5.- I suggest to perform an experimental assays of the most promissory peptide for in silico validation

Response: This phase of the research focuses on designing a universal mRNA vaccine for SARS-CoV-2. Validation and evaluation of the immunogenic properties of the potential vaccine construct will be carried in the next phase of the work.

Round 2

Reviewer 2 Report

Dear Editor

I have revised the article: BIOINFORMATICS DESIGNING AND MOLECULAR MODELLING OF A UNIVERSAL mRNA VACCINE FOR SARS-COV-2 INFECTION, it has been re-phrase improving the article´s content, however, it appears as review and non as an original work. For that reason, I am still suggesting to include the consensus sequences either RNA and for the most promissory peptides submitting these to docking and MD simulations on MHC.  Thus could avoid the experimental assays suggested initially. 

Author Response

Dear Reviewer,

Sequel to mail received on the need to revise and include some information based on the suggestion of the reviewer. As suggested the analysis of the promissory consensus sequences have been included. This is to improve the quality of the manuscript and disprove the fact the study was just a review article.

 However, Molecular docking and molecular simulation with MHC I and II was not carried out, due to the fact the Representation of the Human Leucocytes Antigen in databases is numerous but hope to carried this out subsequently using in vitro approach. The study is to be continued in which that part will be included for further studies.

Furthermore, we intend to validate the vaccine via non-clinical safety evaluation study which is meant to further show case the safety of the vaccine via in vivo means. Also, the releasing of the protein sequence will go against the patency plan for the vaccine candidate.

We hope that this version of the manuscript will be acceptable for publication in your journal.